# The Role of the JC Virus in Central Nervous System Tumorigenesis

**DOI:** 10.3390/ijms21176236

**Published:** 2020-08-28

**Authors:** Nicholas Ahye, Anna Bellizzi, Dana May, Hassen S. Wollebo

**Affiliations:** Center for Neurovirology, Department of Neuroscience, Lewis Katz School of Medicine at Temple University, 3500 N. Broad Street, Philadelphia, PA 19140, USA; nicholas.ahye@tuhs.temple.edu (N.A.); tuf81370@temple.edu (A.B.); tul10237@temple.edu (D.M.)

**Keywords:** Polyomavirus JC, tumors of central nervous system, CY and Mad-4 NCCR-transgenic mice, p53 and pRB oncosuppressor, DNA damage response (DDR), Wnt pathway, insulin receptor substrate-1 IRS-1 signaling

## Abstract

Cancer is the second leading cause of mortality worldwide. The study of DNA tumor-inducing viruses and their oncoproteins as a causative agent in cancer initiation and tumor progression has greatly enhanced our understanding of cancer cell biology. The initiation of oncogenesis is a complex process. Specific gene mutations cause functional changes in the cell that ultimately result in the inability to regulate cell differentiation and proliferation effectively. The human neurotropic Polyomavirus JC (JCV) belongs to the family *Polyomaviridae* and it is the causative agent of progressive multifocal leukoencephalopathy (PML), which is a fatal neurodegenerative disease in an immunosuppressed state. Sero-epidemiological studies have indicated JCV infection is prevalent in the population (85%) and that initial infection usually occurs during childhood. The JC virus has small circular, double-stranded DNA that includes coding sequences for viral early and late proteins. Persistence of the virus in the brain and other tissues, as well as its potential to transform cells, has made it a subject of study for its role in brain tumor development. Earlier observation of malignant astrocytes and oligodendrocytes in PML, as well as glioblastoma formation in non-human primates inoculated with JCV, led to the hypothesis that JCV plays a role in central nervous system (CNS) tumorigenesis. Some studies have reported the presence of both JC viral DNA and its proteins in several primary brain tumor specimens. The discovery of new Polyomaviruses such as the Merkel cell Polyomavirus, which is associated with Merkel cell carcinomas in humans, ignited our interest in the role of the JC virus in CNS tumors. The current evidence known about JCV and its effects, which are sufficient to produce tumors in animal models, suggest it can be a causative factor in central nervous system tumorigenesis. However, there is no clear association between JCV presence in CNS and its ability to initiate CNS cancer and tumor formation in humans. In this review, we will discuss the correlation between JCV and tumorigenesis of CNS in animal models, and we will give an overview of the current evidence for the JC virus’s role in brain tumor formation.

## 1. Introduction

The transformative properties of some viruses and their role in human cancers have been known for many decades. An estimated 10–15% of human cancers worldwide are associated with one of seven known viruses [1,2]. The *Polyomaviridae* family is one salient example of viruses whose known role in the development of human diseases has been evolving [3,4,5]. The JC virus, BK virus, Simian virus 40 (SV40), and more recently discovered Merkel cell Polyomavirus (MCPyV) are some of the members in this group that infect humans and are known to either cause or have an association with human diseases and cancers. These viruses are widespread, but most often only responsible for a clinically silent infection [6,7,8]. Detection of viral nucleic acid or proteins in tumor samples and the study of effects on cellular regulatory pathways have illustrated the mechanisms by which they can transform cells. Persistence of the JC virus in the brain and other human tissues, as well as its oncogenic potential, has made it a subject of study for a role in brain tumor formation [9,10].

Many primary central nervous system tumors remain incurable, even after multimodality treatment with surgical resection, chemotherapy, and radiation therapy. Identification of some key genetic mutations in these tumors that influence growth, affect treatment response, or predict outcomes has increased understanding and influenced new treatment strategies. Yet there remains much to be learned about risk factors and mechanisms of tumorigenesis. Many potential genetic and environmental risk factors for brain tumor formation have been heavily studied, but few have been identified [11]. Discovering the underlying mechanisms driving oncogenesis in the central nervous system can reveal how these malignancies form, grow, and resist treatment, thereby steering the development of more effective therapies. A variety of brain tumors, including glioblastomas [12,13,14,15,16], ependymomas [17,18] medulloblastomas [19,20,21,22,23], astrocytomas and oligoastrocytomas [15,24], and oligodendrogliomas [15,17,25], have been studied for an association with the JC virus. The understanding of the complex role these viruses play in brain tumor formation requires a wide variety of methods. It remains a very active area of research due to the potential benefits of being able to treat better or prevent these cancers if a causative agent is identified. In this review, we present an overview of the current evidence for the JC virus’s role in brain tumor formation.

## 2. History and Epidemiology of JC Virus

The JC virus (JCV) is a human neurotropic Polyomavirus causing progressive multifocal leukoencephalopathy (PML). The JC virus was discovered in 1971 from the brain of a patient with PML [26]. PML lesions have multiple foci of myelin loss, which cause debilitating neurological symptoms. PML lesions are characterized by oligodendrocytes with viral nuclear inclusion bodies and JCV-infected bizarre astrocytes with no signs of apoptosis. The common underlying feature of PML is a severe weakening of the immune system, especially from human immunodeficiency virus (HIV-1)/ Acquired Immunodeficiency Syndrome (AIDS) [27,28,29,30]. PML also occurs in other immunosuppressive conditions such as organ transplantation [31], CD40 ligand deficiency [32], hyper-immunoglobulinemia [33], multiple sclerosis (MS) [34,35], and Crohn’s disease [36].

Due to the asymptomatic nature of JCV infection, it is difficult to pinpoint the period of primary infection. JCV infection occurs in childhood and, several seroprevalence studies have shown 8–10% of children below six years have antibodies against the JC virus [37,38]. Sero-epidemiological data have shown that JCV seropositivity increases during adult life, with rates up to 70% [39,40,41,42,43] or higher of the world’s population having positive titers for JC virus [41,42,43,44]. Based on the structure of the non-coding control region (NCCR), there are two types of JCV variants: Archetype and Mad-1. The Archetype is the most abundant strain of the JC virus in the environment, and it is the transmittable form between individuals [45]. Genotyping analysis based on DNA sequence polymorphism in the C-terminal domain of VP1, intergenic region, and T-antigen gene indicates that JCV transmission occurs within the same family [46,47]. Many different possible routes of infection have been proposed for these viruses, but the definitive answer remains unclear. JCV is most often detected from urine samples of normal and immunosuppressed individuals, indicating urine as a major source of viral transmission [48]. The tonsils and gastrointestinal tract can contain viral DNA and have been suggested as possible initial infection sites [44,49,50,51,52]. After primary infection, JCV is believed to be disseminated by a hematogenous route [53,54,55,56] and maintains in a persistent or latent state in kidneys, bone marrow, and spleen tissue [57,58,59]. From the site of latency, the virus reactivates under immunosuppressed conditions and can reach the brain using B-lymphocytes [60,61]. Although JCV DNA can be detected in different blood cell types, a clear indication of whether the virus undergoes productive infection is missing [62]. It has been reported that the JCV genomic DNA sequence can be found in the brain from immunocompetent patients, implicating the brain as a possible site of latency [63,64,65]. There are many critical areas where our understanding of JCV biology is incomplete. Specifically, the exact JCV latency site, the cell type in which the Archetype form converts to neurotropic strains such as Mad-1, and the mechanism of virus reactivation to cause disease. The finding of both Archetype and neurotropic strains in a different subpopulation of PBMC suggests that the blood compartment might be the site of JCV conversion [66].

## 3. JC Virus Characteristics

The JC virus is a double-stranded DNA virus, with a circular genome contained in a small, non-enveloped icosahedral capsid. The genome consists of early and late coding sequences that are separated by a regulatory region [26,67,68]. The early coding region encodes large T antigen (LTAg) and small t antigen (stAg). In contrast, the viral structural proteins (VP1, VP2, and VP3) and small accessory protein, called Agno, are encoded by the late coding region. The viral structural proteins are essential for early events of the viral life cycle, such as attachment to a cellular receptor, adsorption, and penetration. Between early and late coding regions lies the non-coding control region (NCCR) region, which contains the promoter/enhancer elements for expression of the early and late genes and the origin of viral DNA replication [68,69]. The NCCR region also contains binding sites for a number of transcriptional factors including a unique NF-kB site, C/EBPβ, NFAT4, Rad51, NF-1, SP1, and others [70,71,72,73]. Sequence variation in the NCCR determines JCV tropism and its pathogenic effect [74,75,76,77]. Figure 1 shows a representation of JCV genome with the genes involved in the production of the viral proteins and the NCCR.

The early coding region products are not only involved in DNA replication and transcription, but they are also known to have a role in cellular transformation. The late coding region contains genes that encode for viral capsid proteins, which after assembly, form the infectious virion. The Agno protein is involved in different aspects of the JCV life cycle such as viral replication and transcription [78,79], cell cycle arrest and deregulation [80], viroporin [81,82], and transport of the new virion from the nucleus to the cytoplasm [83,84] without being packaged into the virion structure [85].

The JCV life cycle begins with the interaction of VP1 with α-2,3-linked sialic acid. This binding allows the virus to navigate in the cytoplasm through different cellular compartments to reach the nucleus [86]. Inside the nucleus, viral transcription precedes viral DNA replication since the viral early gene products LTAg, stAg, and splicing variants of LTAg are essential for the initiation and progress of the lytic cycle [87,88]. LTAg of JCV is a multifunctional protein with many domains and is essential for viral DNA replication, where it binds directly to the viral origin of replication. LTAg is responsible for the transcriptional switch from early to late by directly activating late gene expression and downregulating its early promoter [89,90,91]. It has been reported that the splicing variant of LTAg (T prime or T′) cooperates to facilitate LTAg-mediated DNA replication [92]. LTAg modulates many cellular functions through its many domains by interacting with cellular regulators such as pRB and p53 to promote cell cycle progression.

## 4. JCV Early Gene Products and Agno Protein and Their Oncogenic Potential

Several lines of experimental evidence suggest that JC virus infection of primary cells in vitro leads to transformation, and those cells with a transformed phenotype showed the expression of viral early gene products. Although many studies confirm the presence of JC virus in human malignancies, whether there is a direct association of JCV with human cancer remains a topic of debate. The expression of JCV early gene products is strongly associated with the oncogenic potential of JCV, specifically viral LTAg. The LTAg is a nuclear protein with multiple functional domains involved in different viral and cellular functions [93,94].

LTAg is a nuclear phosphoprotein required for the JC virus to replicate its DNA. This protein is ubiquitous amongst members of the Polyomavirus family. After viral infection and the beginning of viral genome replication, LTAg protein interacts with the transcription origin. It also can promote progression of the cell life cycle into S-phase by a variety of regulatory protein interactions. This step is necessary for the completion of the viral life cycle. These significant protein interactions include retinoblastoma protein (pRb) [95,96] and the tumor suppressor p53. The interaction of LTAg with pRb leads to the activation of cellular elongation factors to promote cell cycle progression [97]. Normally, pRb sequesters the E2F transcription factor and prevents cell cycle progression from G1 to S phase. Inactivation of pRb by LTAg binding releases E2F and promotes cellular proliferation [95,98,99]. Besides, LTAg interacts with p53 [100,101] to inhibit DNA repair and apoptosis. Briefly, the release of E2F from pRb by LTAg activates p14^ARF^ expression, which leads to the stabilization of p53. However, LTAg binds and inactivates p53, preventing the p53 action in response to the DNA damage or p14^ARF^ production [99]. In mice transgenic for JCV early region, LTAg may also inhibit the tumor suppressor activity of p53 through the interaction with merlin, a product of the gene neurofibromatosis type 2 (NF-2), that neutralizes the inhibitory effect of Mdm2 on p53 [102]. The disease NF-2 is characterized by an NF2 mutation resulting in the development of tumors which histologically resemble malignant peripheral nerve sheath tumors [103,104] (Figure 2 and Table 1).

LTAg is also known to interact with components of different signaling pathways which are associated with cellular transformation such as β-catenin [3,105], insulin receptor substrate -1 (IRS-1) [106], and survivin [107].

β-catenin is a crucial protein of the Wnt signaling pathway, usually located and degraded in the cytoplasm. The central domain of LTAg can bind the C-terminus of this protein [105], resulting in its nuclear translocation, and subsequent activation of c-Myc and cyclin D1 TCF promoter, leading to cellular proliferation [108]. These signaling events occur in human cancers associated with JCV, including medulloblastoma, colon cancer, and esophageal cancer [3,109,110,111]. LTAg can also stabilize β-catenin through a non-canonical Wnt signal pathway, recruiting the GTPase protein Rac1. Rac1 stabilizes β-catenin and prevents its degradation by ubiquitin-mediated proteasome, allowing its nuclear translocation [112] (Figure 2 and Table 1).

IRS-1 is the downstream docking molecule of the insulin growth factor 1 receptor (IGF-1R) pathway. As for β-catenin, LTAg is stabilized by binding IRS-1 in the cytoplasm with the result of its nuclear translocation [106]. The unusual presence of IRS-1 in the nucleus enhances its binding and inactivation of enzyme Rad51, which is involved in repairing DNA double-stranded breaks (DSBs) by homologous recombination (HR). HR is a high-fidelity DNA repairing process characterized by a significant amount of energy and an active cell division stage during which a homologous DNA template is available. However, if the HR result is compromised, damaged cells are forced into non-homologous end-joining (NHEJ) recombination, a primitive process in which the loose ends of the DNA breakage are simply rejoined without the involvement of a homologous template, resulting in accumulation of mutations. The inactivation of Rad51 by nuclear translocation of LTAg-stabilized IRS-1 prevents HR, forcing the cell to repair its DSBs via NHEJ [113] (Figure 2 and Table 1).

LTAg cooperation with IGF-1R is linked to an increased level of survivin, a potent anti-apoptotic protein normally expressed during development, but completely silenced in a differentiated tissue. The in vitro induction of LTAg expression in wild type IGF-1R neural progenitors increased the survivin expression threefold and accelerated cell proliferation. In contrast, in IGF-1R-knockout neural progenitors, LTAg expression failed to increase survivin expression, resulting in massive apoptosis induction [114]. It has also been shown that LTAg may activate the survivin promoter, and infection of glial cell cultures with JCV resulted in a significant expression of survivin, which protected infected cells from apoptosis [107] (Figure 2 and Table 1).

Infection of glial cells by JCV also inflicts severe DNA damage on host cell DNA, which is manifested by an increase in ploidy, micronuclei formation, and expression of γH2AX, all of which are indicative of DNA damage and involve the viral early transforming protein LTAg [115]. The first indication of an association between Polyomavirus infection and chromosomal damage was reported by Lazutka JR et al. [116]. They showed the correlation between high titers of JCV and BKV antibodies with increased frequency of chromosomal damage in human lymphocytes. The ability of LTAg to influence cell cycle regulation and DNA transcription and repair is part of the repertoire of transformative effects on an infected cell. It has been recently reported that there is a direct association of Polyomavirus BK in some urothelial neoplasms arising from kidney transplantation. In this case, reports suggest that BK virus integrated into human chromosomes in tumor cells with no productive infection but with robust expression of LTAg. This dysregulation of LTAg expression in non-lytic cells might drive cell growth, DNA damage, and tumorigenesis [117]. In the case of JCV, we and others have reported the activation of DNA damage response during JCV infection or transient expression of LTAg which is associated with activation of ataxia-telangiectasia mutated (ATM) and ATM- and Rad3-Related (ATR) kinases [118,119]. This molecular interaction of JCV with the components of the DDR facilitate conditions that promote viral replication at the cost of host genomic instability that may lead to tumorigenicity [120]. The induction of the DDR by infection may be a general feature of Polyomaviruses [121]. The two human Polyomaviruses BKV and MCPyV also induce the DDR through activation of ATM and ATR kinases. However, these viruses utilize DDR not only to promote their viral replication but also to cause carcinogenesis at the expense of host DNA damage [117,122,123,124,125] (Figure 2 and Table 1).

Small t antigen (stAg) is another significant protein that has roles in viral production and influencing host cell growth. Research on stAg has revealed its oncogenic effects [126]. Its interactions with specific protein phosphatases, which generally suppress cellular growth pathways, permits increased activation of those pathways. These include retinoblastoma proteins and protein phosphatase 2A (PP2A), both of which have host cell functions that can lead to transformation when inhibited [78,126,127,128,129]. PP2A is a serine/threonine phosphatase that regulates phosphorylation signals activated by kinases, such as mitogen-activated protein kinase (MAPK), that promote cell proliferation. stAg can interact with PP2A to regulate DNA repair mechanisms [130], inhibition of the Wnt signaling pathway, and alteration in cytoskeletal proteins and tight junctions which increases invasiveness [131]. Recently it has also been shown that MCPyV small t antigen is an oncoprotein that can transform rodent fibroblast in vitro independent of LTAg [132,133]. Small t is an essential enhancer of cell proliferation in MCC [134] and in vivo [135] (Figure 2 and Table 1).

Finally, it seems that the expression of the late JCV protein Agno also affects the DDR. The cells expressing Agno were more sensitive to the cytotoxic effects of cisplatin with consequent increased chromosome fragmentation, micronuclei formation, and cell cycle impairment [136]. NHEJ was impaired in nuclear extracts from cells that expressed Agno. It has been hypothesized that defects in NHEJ were caused by binding of Agno protein to the Ku70 DNA repair protein and subsequent sequestering in the perinuclear space [115]. Moreover, in mouse fibroblasts constitutively expressing Agnoprotein, an increased level of p21/WAF-1 protein was observed with dysregulation of cell cycle progression, and an accumulation of cells in the G2/M phase. Besides, activation of p21/WAF-1 gene expression in these cells is partly mediated through the cooperation of Agno protein with p53 [80] (Figure 2 and Table 1).

## 5. JCV Oncogenicity in Animal Models

While the tumorigenic potential of polyomaviruses such as JCV and BKV in humans is still a matter of debate (see Table 1), the JCV tumorigenic role in animal models is well documented. The first evidence of the association of JC virus with cancer was reported with the development of different brain tumors within 3–12 months when newborn golden Syrian hamsters were inoculated with JCV. The majority of the tumors were glial origins, such as medulloblastomas that originate in the cerebellum [137,138] and gliomas that originate in the brain [139]. Neuroblastoma, a solid embryonal tumor of the sympathetic nervous system arising from the neural crest, was also reported [140]. The kind of tumors that developed in hamsters was strictly related to the site of injection and the JCV strain. JCV Mad-1 strain isolated from a PML patient was able to induce malignant gliomas within six months in 83% of newborn hamsters inoculated intracerebrally and subcutaneously [139]. Moreover, JCV from hamster tumor cells was capable of replicating after the fusion of these cells with one permissive to the viral infection, confirming the persistence of the JCV genome in cultured tumor cells [139]. Another study came to the same conclusion as reported by Walker and coworkers [138]. Additional studies have further confirmed the neuro-oncogenicity of other strains of JCV in newborn hamsters. The 95% of hamsters inoculated with JCV Mad-2 developed cerebellar medulloblastomas similar to those inoculated with the Mad-1 strain. In contrast, inoculation of the Mad-4 strain resulted in 45% of animals with pineal gland tumors and 45% of animals with tumors in the cerebellum [137] Although different strains of JCV were associated with the induction of various brain tumors, the route of JCV inoculation has also been shown to correlate with the kind of tumor observed in newborn hamsters. As reported by Varakis and colleagues, hamsters administered with JCV Mad-1 through the eye developed neuroblastomas after 6–11 months, and primary tumors also arose in the abdominal cavity with metastasis in several organs [140] (Table 2). Finally, there was no direct evidence of viral replication or detection of viral structural proteins in all studies conducted to date in tumor tissues [138,139,141,142,143,144,145].

The pioneering studies in hamsters prompted the researchers to focus on the association between JCV and brain tumors in non-human primates. The first studies have reported that intracranial injection of JC virus into owl monkeys and squirrel monkeys led to the development of astrocytoma within 14–36 months [142,146]. Analysis of the specimens from monkey tumors showed the presence of LTAg in tumor tissue without active viral replication, as evident from the lack of detection of the viral structural proteins [142] (Table 2). Interestingly, a suspension of tumor tissue isolated from owl monkeys with JCV-induced astrocytoma [142] was successfully maintained in culture and analyzed for phenotypic changes generally associated with expression of the LTAg [147]. JCV LTAg appeared in the nucleus of these cultured cells, not in complex with the onco-suppressor p53 but in the presence of actin dealignment features. These observations led the authors to conclude that JCV LTAg expression can persist in brain tumor cells and correlate with cell phenotypes typical of Polyomaviruses-related transformation, with tumor development in monkeys visible only several years after the first viral inoculation [147].

Further support of the role of JC virus in tumor development was reported by Ohsumi’s laboratory that showed the induction of neuroectodermal tumor after Tokio-1 JCV strain injection into the brains of newborn Sprague–Dawley rats [148,149,150] (Table 2). However, compelling evidence linking JCV with oncogenicity comes from a transgenic mice study. These transgenic mice were generated by inserting the entire gene for JCV LTAg under the control of its promoter expressing only LTAg. In particular, Krynska and colleagues generated transgenic mice containing the early region of the archetype strain CY. These mice exhibited paralysis of rear limbs, poor grooming, and hunched posture within 9 and 13 months of age. The autopsy revealed the presence of primitive tumors from the cerebellum, resembling the human medulloblastoma/primitive neuroectodermal tumor (PNETs). Moreover, LTAg was expressed in the nuclei of all tumor tissue analyzed by immunohistochemistry, strongly suggesting the potential of this animal model for the study of human CNS tumors in association with the Polyomavirus JCV [151] (Table 2). A subsequent study in transgenic mice expressing LTAg under the promoter region of JCV Mad-4 strain developed large pituitary neoplasia within one year in 50% of animals. The same study also demonstrated the interaction of LTAg with p53 with the overexpression of the p53-downstream target protein p21/WAF1. These results have contributed to make this animal model a useful tool to evaluate further mechanisms of tumorigenesis mediated by JCV LTAg [152] (Table 2). As observed in hamsters, it is essential to underlay that the type of malignant transformations developed in murine models also seems to be dependent on the strain of JCV [151,152,153,154]. All in all, these findings suggested the tumorigenic role of JCV LTAg.
ijms-21-06236-t001_Table 1Table 1JCV-related Oncogenic Mechanism.Signaling PathwayCellular FactorJCV FactorOncogenic EffectReferencesTumor suppressorsp53, p21^WAF1^Agnocell cycle arrest in G2/M in vitroDarbinyan A et al. 2002 [80]p53, p21^WAF1^LTAgpituitary neoplasia in LTAG transgenic mice Gordon J et al. 2000 [152]pRbLTAgcell cycle progression in vitroDyson N et al. 1990 [95]pRb2/p130, E2F4/5 LTAgcell cycle progression in vitroCaracciolo V et al. 2007 [97]p53, pRbLTAgcell cycle dysregulation in tumor formation in LTAg transgenic mice Krynska B et al. 1997 [99]NF2LTAgtransgenic mouse model of malignant peripheral nerve sheath tumorsShollar D et al. 2004 [103]pRb, PP2AstAgcell cycle dysregulation and viral DNA replicationBollag B et al. 2010 [127]; Sariyer IK et al. 2008 [128]; Pallas DC et al. 1990 [129] Wnt-catenin, c-Myc, Cyclin D1LTAgoncogenesis of colon cancerEnam S et al. 2002 [3]; Ripple MJ et al. 2014 [108]-cateninLTAgmouse medulloblastoma cell line (BSB8), JCV-induced hamster astrocytoma cell line (HJC2) and human astrocytoma U-87MG cell lineGan DD and Khalili K 2004 [105]-catenin, LEF-1/TCF promoterLTAgmurine medulloblastoma cell line (BsB8)Gan DD et al. 2001 [111]Rac1 GTPaseLTAg-catenin stabilization and cell cycle progression in vitroBhattacheryya R et al. 2007 [112]PP2AstAgInhibition of Wnt signaling, alteration in cytoskeleton proteins and increase of invasivenessNunbhakdi-Craig V et al. 2003 [131]IGF-1RIRS-1LTAgtranslocation to the nucleus and cell cycle progressionLassal A et al. 2002 [106]survivinLTAgapoptosis inhibition Piña-Oviedo S et al. 2007 [107]survivinLTAgapoptosis inhibition and proliferation of neural progenitorsGualco E et al. 2010 [114]IGF-1R and DDRIRS-1, Rad51LTAgHR dysregulation and DNA damageTrojanek J et al. 2006 [113]DDRNHEJ Ku70AgnoHR dysregulation and DNA damageDarbinyan A et al. 2004 [136]HR Rad51, NHEJ Ku70, H2AX LTAg, AgnoHR dysregulation and DNA damage (mutation, ploidy, and micronuclei formation)Darbinyan A et al. 2007 [115]HR Rad51, ATMLTAgDNA damageWhite MK et al. 2014 [73]; White MK et al. 2017 [118]PP2AstAgDNA damageHuang JL et al. 2015 [130]HR: Homologous recombination; NHEJ: Non homologous end join.
ijms-21-06236-t002_Table 2Table 2JCV oncogenesis in animal model.Animal ModelJCV DeliveryTumorsAssayReferencesGolden Syrian Hamsters(*Mesocricetus auratus*)newborns inoculated intracerebrally and subcutaneously with JCV isolated from a patient with PMLmalignant gliomas: most of the tumors were glioblastomas and medulloblastomastransplantation of tumors subcutaneously and isolation of JCV from 5/7 tumors tested. Cells from four of these tumors were cultivated in vitro: intranuclear LTAg antigenically related to SV40 LTAg; JCV virions after fusion of this culture with permissive cellsWalker DL et al. 1973 [139]three groups of newborns inoculated intracerebrally with three different JCV strains (Mad-2, Mad-3, and Mad-4) cerebellar medulloblastomas with Mad-2 inoculation; pineal gland tumors and tumors in the cerebellum with Mad-4 inoculation.histologic characterization of tumors.Padgett BL et al. 1977b [137]one group of newborns inoculated intraocularly. Another group was inoculated subcutaneously and intraperitoneally. Both with JCV Mad-1 strainneuroblastomas and primary tumors in the abdominal cavity with metastasis in liver, bone marrow, and lymph nodes.two neuroblastomas were transplanted serially, and a tissue culture cell line was established from one of them. T-antigen was detected in 3/5 primary tumors tested and in the transplanted tumors.Varakis J et al. 1978 [140]newborns inoculated intracerebrally and subcutaneously with JCV isolated from a patient with PMLmedulloblastoma involved the internal granular layer of the cerebellum: lesion comparable to childhood human medulloblastoma LTAg IF and histologyZuRhein GM et al. 1979 [138]newborns inoculated intracerebrally with Tokio-1 JCV strain (isolated form a patient with PML, serologically identical to Mad-1 strain).cerebellar medulloblastomaLTAg IF and histology (Homer-Wright rosettes)Nagashima K et al. 1984 [148]Owl Monkeys(*Aotus trivirgatus*)two animals inoculated intracerebrally, subcutaneously, and intravenously with JCV isolated from a patient with PMLastrocytoma (resembling human glioblastoma multiforme) and a malignant tumor containing both glial and neuronal cellsTAg IF and histologyLondon WT et al. 1978 [142]Squirrel Monkeys(*Saimiri sciureus*)six animals inoculated intracerebrally, subcutaneously, and intravenously with JCV isolated from a patient with PMLastrocytomas in 4/6 animals. histologic characterization of tumorsLondon WT et al. 1983 [146]Sprague-DawleyRatsnewborns inoculated intracranially with Tokyo-1 JCV strain.brain tumors in the cerebrum: undifferentiated neuroectodermal nature and pseudo-rosettes. LTAg IHC and histology. Neuronal differentiation was not proved. Glial differentiation was confirmed by subcutaneous transplantation of cultured tumor cellsOshumi et al. 1985 [149]; Oshumi et al. 1986 [150].Transgenic Micetransgenic mice for the early region of JCV Archetype strainprimitive tumors originating from the cerebellum: close resemblance of human medulloblastoma/primitive neuroectodermal tumors (PNETs)RT-PCR for LTAg mRNA, IHC for LTAg and p53, IP for LTAg and p53 and Archetype NCCR sequencingKrynska B et al. 1999b [151]transgenic mice for the early region of JCV Mad-4 strainpituitary neoplasiaIHC for LTAg and p53, IP for LTAg, p53 and p21^WAF1^Gordon J et al. 2000 [152]transgenic mice for the early region of JCV Mad-4 strainpituitary neoplasia and signs resembling malignant peripheral nerve sheath tumors.IHC for LTAg, NF-1, NF2,p53, and p21^WAF1^ and IP for LTAg, NF-1, NF2 and p53,Shollar D et al. 2004 [103]


## 6. Evidence of JCV Infectivity in Human Tumor Tissues

The direct link of JCV with human cancer is still a matter of debate. JCV causes tumors in rodents and non-human primates, which are not its natural host. The JCV ubiquitous nature in the population makes it hard to establish its role in human cancer. However, there have been reports that indicate the expression of the viral LTAg in association with the transformation of neuronal cells in vitro and the induction of tumors in monkeys, which lead to speculation of the possible association of JCV with human CNS tumors. The first evidence linking the JCV association with a human brain tumor (oligodendroglioma) was reported in a patient with chronic lymphocytic leukemia with PML [155]. JCV particles were detected in multiple gliomas and multiple and malignant astrocytomas in patient with characteristic PML lesions [156,157]. However, the first direct evidence implicating JCV and its viral protein in CNS neoplasms was described in a 21-year-old patient with immunodeficiency and PML. *Postmortem* examination of brain tissue by immunohistochemistry and in situ hybridization showed the expression of viral LTAg and mRNA, respectively [158]. The association of JCV with CNS tumors in animal models prompted the undertaking of a large-scale analysis of human brain tumor tissue samples for the presence of viral DNA or proteins [159]. The first systematic analysis of brain tumor tissues for the detection of JCV DNA and proteins was conducted in 1996 by Rencic and colleagues [24] in a patient with an oligoastrocytoma in the absence of PML. In this study, the presence of JCV DNA was confirmed by sequencing of the PCR products. JCV RNA and T-Ag protein were detected in the tumor tissue by primer extension analysis and Western blotting, respectively. Del Valle and colleagues [15] examined 85 samples of glial tumors for the presence of JCV DNA sequences and T-Ag expression, showing that 57% to 83% of tumors were positive for JCV.

Further studies by the same Authors linked JCV to other brain tumors, such as glioblastoma multiforme, oligoastrocytoma, oligodendroglioma, and medulloblastoma (Table 3). In particular, medulloblastoma is among the most frequent grade IV brain tumor with the highest number of cases in children of age between six and eight years. Histologically and morphologically, medulloblastomas are embryonal tumors derived from neuronal stem cells of the cerebellum [160]. Krynska and colleagues investigated the association between JCV and pediatric tumors in humans. They showed the presence of JCV DNA encoding N-terminal and C-terminal of LTAg in 11 out of 23 pediatric medulloblastoma tissue samples [20]. They hypothesized the “*hit*-*and*-*run*” mechanism where LTAg expression may trigger a cascade of tumorigenic events that do not require the presence of the viral protein in the advanced stage of tumoral progression. In support of these outcomes, earlier studies by the same research group in a transgenic murine model of medulloblastomas induced by JCV early gene expression showed that not all the tumoral cells produced LTAg [151]. Moreover, the N-terminus region of JCV LTAg that interacts with the onco-suppressor pRb was detected in 87% of the pediatric medulloblastomas. In contrast, the C-terminal region was detected in only 56.5% of the samples. The finding of a particular critical domain of LTAg, which is known to interact with onco-suppressor proteins, might implicate Tag to play a role in this specific tumor [20].

In a later study by the same group, 20 paraffin-embedded well-characterized medulloblastomas were analyzed for the presence of JCV Agno DNA sequences and the expression of Agnoprotein and LTAg. PCR analysis revealed the presence of JCV Agno DNA in 11 of 16 samples analyzed, whereas Agnoprotein was detected by immunohistochemistry in the cytoplasm of neoplastic cells in 11 of 20 tissue samples. Moreover, LTAg protein was reported in the nucleus of neoplastic cells where agnoprotein was absent. Finally, none of the tumor samples analyzed has shown the expression of viral late capsid proteins, excluding the possibility of a productive JCV infection of the tumor cells. These data provided additional evidence that both LTAg and Agno might play a role in the development of JCV-associated medulloblastomas [22].

Further evidence for the possible role of LTAg and Agno in GBM was reported in two cases having GBM. The first case is an immunocompromised individual with multiple sclerosis, and the second one is a 54-year-old immunocompetent patient. In the first case study, GBM tumor tissues were positive for JCV DNA, and LTAg was localized in the nucleus. The same conclusion was reported in the second case with the additional evidence of Agno protein expression in the cytoplasm. Furthermore, using laser capture microdissection techniques, Mad-1 strain was identified in the tissues examined [16,23] (Table 3).

Finally, since it has been reported that JCV can infect B-lymphocytes, it leads to speculation of its association with primary CNS lymphomas. Del Valle et al. in 2004 published a study involving 27 cases of CNS lymphomas, in which the JCV DNA sequence was detected in 81% of cases, but the expression of LTAg was detected in only 18.5% [161]. Several studies have shown the coexistence of PML and primary CNS lymphomas indicating PML-associated JCV reactivation [162,163]. JCV infection of B-lymphocytes in vitro is non-productive, which leads to the speculation that infected B-lymphocytes may serve as a carrier to disseminate the virus to the CNS, where it can infect glial cells [61] (Table 3). The spread of JC virus from the initial site of infection to the brain is one of the many critical areas where our understanding of JCV biology is incomplete. Specifically, the mechanism by which JCV reaches the brain from the primary peripheral site of infection. It has been hypothesized that immune cells, specifically the B-lymphocytes, not only serve as latent reservoirs [28,164,165,166] but also an agent disseminating the virus to CNS-specific glial cells.

## 7. Conclusions

Multiple experiments have identified the oncogenic potential of the JC virus: its transformative ability of cells in vitro and in in vivo models. Several studies reported the detection of viral genome and proteins by a molecular and virologic approach in many organs and tissues, including the brain. The oncogenic potential of the human BKV with viral genome integration into the cellular DNA has recently been shown. Using high-through sequencing of tumor DNA obtained from urothelial carcinoma, the researchers identified the integration site of the BKV genome into exon 26 of myosin-binding protein C1 gene (MYBPC1) on chromosome 12 in tumor cells but not in normal renal cells. Interestingly, this viral integration event leads to altered viral gene expressions such as robust expression of LTAg and a lack of expression of the viral structural proteins and DNA replication. This finding supports the notion that a polyomavirus integration event is essential to tumorigenesis [117]. In the case of BKV, the persistent over-expression of LTAg in non-lytic cells likely promotes cell growth, genetic instability, and oncogenesis [122]. Furthermore, the viral integration event has been considered a critical step in MCPyV-mediated tumorigenesis. Several studies have shown the clonal integration of MCPyV into Merkel cell carcinoma, which leads to persistent expression of the oncoproteins LTAg and stAg [167]. Although the integration of JCV has been reported in an in vitro model, there is still no definitive evidence that shows JCV integration into host genome as responsible for tumors in humans. 

The finding of JC virus DNA and its proteins, in either normal or brain tumor tissues, establishes an active presence in the brain [63]. There are conflicting reports regarding the quantities of JC viral DNA found in human tumor specments [168,169,170]. However, the inability to detect the viral DNA sequence does not negate its ability to transform a cell. A more recent concept in tumor virology suggests that the initial infection produces changes and conditions that promote host cell transformation. A viral genome can be inserted into that of the host cell, and it could be lost over time. Despite this, the ability of JC viral proteins to affect the cell cycle and disrupt DNA repair pathways could have long-lasting changes in a cell without the active virus needed to be continually present. While not all tumor samples contain it, viral DNA lasting effects could still be enough to transform the host. Abortive JC virus infection of cells may lead to the expression of viral oncogenic proteins that may initiate tumorigenesis, even if the entire virus is no longer active. 

Since the discovery of Rous sarcoma virus (RSV) as a causative agent of chicken tumor, the direct link of viruses with cancer is now well accepted; 12–20% of human cancers are reported to be caused by a viral infection. The association of human cancer with the infection of some DNA or RNA viruses such as Hepatitis B virus [171], Merkel Cell Carcinoma virus [172], Epstein Barr virus (EBV) [173], Hepatitis C virus [174], Human Herpesvirus Type 8 [175], human papillomaviruses [176] human BKV [117], and the human T-cell lymphotropic virus (HTLV) [177] is well documented. Although JCV has a very high prevalence of infection in the worldwide population, as demonstrated by sero-epidemiological data, there is not yet a proportional number of cancers attributable to this virus. This observation suggests there are multiple immune system defenses or pathways that revert potential virulent effects. Further research is warranted to shed light on all those cellular pathways involved in JCV-associated oncogenesis.

## Figures and Tables

**Figure 1 ijms-21-06236-f001:**
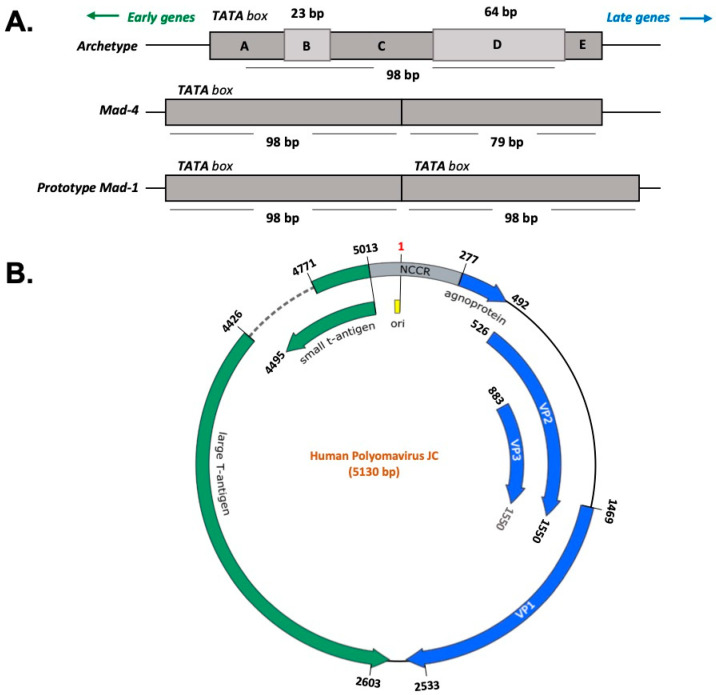
Polyomavirus JC genome. (**A**) The Non-Coding Control Region (NCCR) is the most variable region of the JCV genome and determines the viral strains. The Prototype Mad-1 strain is characterized by a sequence of 98 bp repeated *in tandem*. Mad-4 differentiates from Mad-1 in the deletion of 19 base pairs in the second 98 bp repeat. The Archetype strain is composed of a single sequence of 98 bp with two insertions of 24 bp and 64 bp. (**B**) The NCCR is located between the two coding regions of the JCV genome: the early and the late regions. The early region encodes for the large T antigen (LTAg) and the small t antigen (stAg), whereas the late region contains the genes for the Agnoprotein and the capsid proteins VP-1, VP-2, and VP-3. The numbering of the nucleotide positions refers to the prototype Mad-1 strain (NCBI Reference Sequence: NC_001699.1).

**Figure 2 ijms-21-06236-f002:**
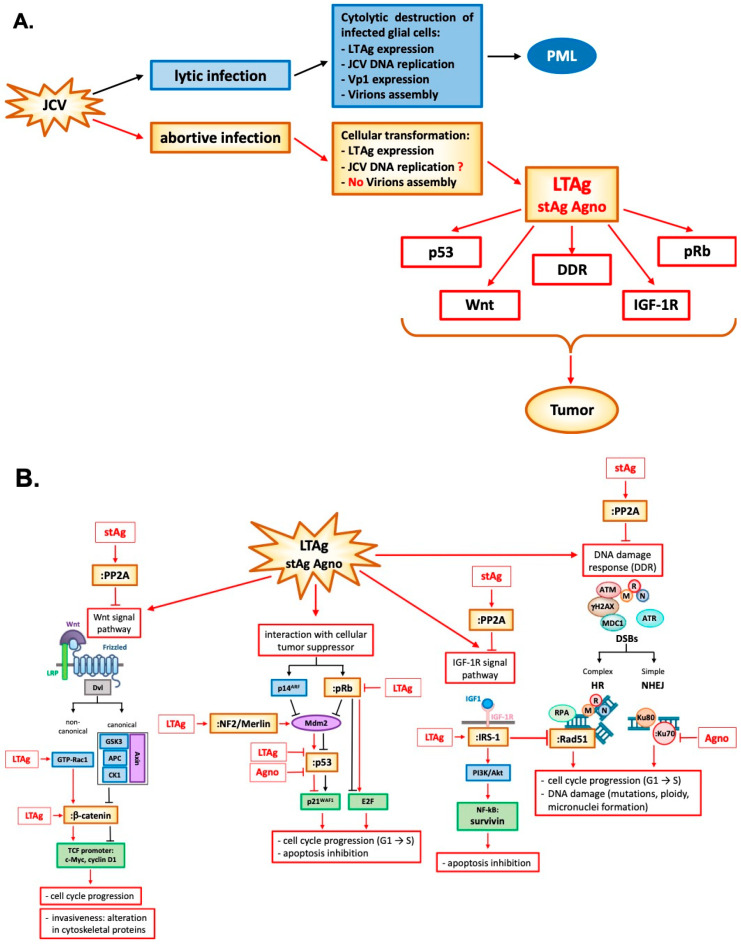
Polyomavirus JC: one virus, two stories. (**A**) In cells permissive to JCV infection, such as oligodendrocytes, viral transcription proceeds the viral DNA replication since the product of the viral early genes LTAg and stAg is essential for initiation and progression of the lytic cycle in oligodendrocytes and progression of PML. JCV LTAg is a multifunctional protein with many domains and it is important for viral DNA replication and for the transcriptional switch from early to late genes which culminates in the production of capsid proteins VP1, VP2, and VP3 and final virions assembly. In cells non permissive to the viral infection, LTAg protein starts modulating many cellular functions through its many domains by interacting with cellular regulators, such as pRb, p53, Wnt, and IGF-1R signaling pathways and DNA damage response factors, promoting cell cycle progression, apoptosis inhibition and DNA damage which culminate in tumor onset. (**B**) In detail, the inactivation of pRb by LTAg promotes cell cycle progression through the release of E2F and activation of p14^ARF^ expression which leads to the stabilization of p53. However, LTAg binds and inactivates p53 preventing the p53 action in response to the DNA damage or p14^ARF^ production. In mice transgenic for the JCV early region, LTAg may also inhibit the tumor suppressor activity of p53 through the interaction with neurofibromatosis type 2 (NF-2), a protein that neutralizes the inhibitory effect of Mdm2 on p53, with the development of tumors. LTAg is also known to interact with components of different signaling pathways which are associated with cellular transformation such as β-catenin, insulin receptor substrate -1 (IRS-1), and survivin. β-catenin is a crucial protein of the Wnt signaling pathway normally located and degraded in the cytoplasm. LTAg can bind the C-terminus of this protein, resulting in its nuclear translocation and subsequent activation of c-Myc and cyclin D1 TCF promoter, leading to cellular proliferation. LTAg can also stabilize β-catenin through a non-canonical Wnt signal pathway, recruiting the GTPase protein Rac1 that stabilizes β-catenin by inhibiting its ubiquitin-dependent proteasomal degradation. IRS-1 is the downstream docking molecule of the insulin growth factor 1 receptor (IGF-1R) pathway. As for β-catenin, LTAg binds and stabilizes g IRS-1 in the cytoplasm with the result of its nuclear translocation. The unusual presence of IRS-1 in the nucleus enhances its binding and inactivation of enzyme Rad51 which is involved in repairing of DNA double-stranded breaks (DSBs) by homologous recombination (HR). The inactivation of Rad51 prevents the HR forcing the cell to repair its DSBs via non-homologous end-joining (NHEJ). LTAg cooperates also with IGF-1R increasing the level of survivin, which protects infected cells from apoptosis by dysregulation of cellular homeostasis and oncogenesis. Infection of glial cells by JCV inflicts also severe cellular DNA damage throughout LTAg which inactivates the ataxia-telangiectasia-mutated (ATM) and ATM- and Rad3-Related (ATR) kinases. Small t antigen (stAg) is another significant viral protein that has roles in viral production and influencing host cell growth. Its interactions with retinoblastoma proteins (pRbs) and protein phosphatase 2A (PP2A) result in alterations of DNA damage response, inhibition of the Wnt signaling pathway, and alteration in cytoskeletal proteins. Finally, Agno protein also affects the DDR: Agno protein can bind to the Ku70 DNA repair protein, sequestering it in the perinuclear space and impairing the NHEJ. In addition, the cooperation between Agno protein and p53 seems to induce the activation of p21/WAF-1 gene expression.

**Table 3 ijms-21-06236-t003:** Association between JCV and human brain tumors in the absence of PML.

Brain Tumor	JCV Factor	Cellular Factor	Assay	References
Glioblastoma	VP1, NCCR	-	PCR and sequencing (Mad-4 NCCR and genotype1 VP1)	Boldorini R et al. 2003 [12]; Delbue S et al. 2005 [13]
	LTAg	p53	IHC (p53 and LTAg), PCR (LTAg) and SB (LTAg)	Del Valle et al. 2000 [14], Del Valle et al. 2001a [15]
	LTAg, VP1, Agno, NCCR	p53	IHC (p53 and LTAg–VP1 not detected), PCR (LTAg, VP1, Agno, NCCR), SB (LTAg, VP1, Agno, NCCR), sequencing (Mad-1NCCR) and LCM LTAg positive cells	Piña-Oviedo S et al. 2006 (case report) [16];
	LTAg, VP1, Agno, NCCR	p53	IHC (p53 and LTAg–VP1 not detected), PCR (LTAg, VP1, Agno, NCCR), SB (LTAg, VP1, Agno), sequencing (Mad-4 NCCR)	Del Valle L et al. 2002b (case report) [23]
Astrocytoma	LTAg	p53	IHC (p53 and LTAg), PCR (LTAg) and SB (LTAg)	Del Valle et al. 2001a [15]
	LTAg, NCCR	-	IHC (LTAg), PCR (LTAg and NCCR) and sequencing (Mad-4 NCCR)	Caldarelli-Stefano R et al. 2000 [17]
Oligoastrocytoma	LTAg	p53	IHC (p53 and LTAg), PCR (LTAg) and SB (LTAg)	Del Valle et al. 2001a [15]
	LTAg, NCCR	Ki67	IHC (Ki67 proliferation marker and LTAg), PCR (LTAg and NCCR), SB (LTAg), primer extension (LTAg), IP (LTAg) and sequencing (Mad-4 NCCR)	Rencic A et al. 1996 [24]
Oligodendroglioma	LTAg	p53	IHC (p53 and LTAg), PCR (LTAg) and SB (LTAg)	Del Valle et al. 2001a [15]
	LTAg, VP1, Agno, NCCR	p53	IHC (p53, LTAg, Agno–Vp1 not detected), PCR (LTAg, VP1, Agno, NCCR), SB (LTAg, VP1 and Agno), sequencing (Mad-4 and Archetype NCCR)	Del Valle et al. 2002c [25]
Ependymoma	LTAg	p53	IHC (p53 and LTAg), PCR (LTAg) and SB (LTAg)	Del Valle et al. 2001a [15]
Medulloblastoma	LTAg, VP1	-	IHC (LTAg–VP1 not detected), PCR (LTAg, VP1), SB (LTAg, VP1)	Krynska B et al. 1999a [20]
	LTAg, VP1	p53, pRb (p107, pRb2/p130)	IHC (p53, pRb, LTAg), PCR (LTAg, VP1)	Del Valle et al. 2001c [21]
	LTAg, Agno	p53	IHC (p53, LTAg and Agno), PCR (LTAg, Agno), SB (LTAg, Agno)	Del Valle et al. 2002a [22]
Primary CNS lymphoma	LTAg, Agno, VP1	p53	IHC (p53 and LTAg–VP1 not detected), PCR (LTAg, VP1, Agno), SB (LTAg, VP1, Agno), LCM LTAg positive cells	Del Valle et al. 2004 [161]

IHC: Immunohistochemistry; SB: Southern Blot; LCM: Laser capture microdissection.

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
