# Peer review of "The Role of the JC Virus in Central Nervous System Tumorigenesis"

_ijms, 2020, doi:10.3390/ijms21176236_

Round 1
Reviewer 1 Report
This is a very nice and thorough review on JCV and CNS oncogenesis written by one of the leading laboratories on this topic. I particularly liked the summarized tables separating all different aspects of JCV interaction with multiple proteins, pathways, etc.
There were only occasional typographical errors that should be checked:
Pg 4, 2nd paragraph, line 3: should read "antibodies againts JC".
Pg 4, 2nd paragraph, line 8: Sentence "Genotyping analysis based... within the family" is confusing, please clarify.
Pg 7, 2nd paragraph, line 3: "c-Myc and cyclin D1 TCF promoter..." there may be a missing comma.
Pg 8, 1st paragraph, line 8: "In this case, it reported..." is confusing, please clarify.
Pg 9, 1st paragraph, line 6: should read "JCV with human CNS tumors". Same page last sentence, "showing that 57% to 83% of tumors..." should it include the word "respectively", please add if needed.
Pg. 10, 1st paragraph, line 1: authors instead of Authors.
Pg 12, 2nd paragraph, last line: monkey should read "monkeys"
Please make sure genes are italicized appropriately in all text. Also double check abbreviations as some of this are not followed in the text. Also, review name of proteins, "Tag" instead of TAg in few instances. Maintain consistency either as all sentences saying B-lymphocytes or B lymphocytes.
Conclusion needs to be shortened. It also sounds repetitive from what has been mentioned already in the text.
If not submitted, please allow to review submitted figures.
Thank you.
Author Response
Manuscript ID: ijms-857066.
The role of JC Virus in central Nervous System Tumorigenesis
Journal: IJMS
Corresponding Author: Hassen S. Wollebo
Date 8/22/20
We thank the editor and reviewers for reading our manuscript and providing helpful comments. Our detailed response is included below. Appropriate changes have been made to the text of the manuscript to address the points that were raised and incorporated into the revised manuscript.
Dear Dr. Wollebo,
Reviewers have now commented on your paper. You will see that they are advising that you revise your manuscript.
When revising your work, please submit a list of changes or a rebuttal against each point, which is being raised when you submit the revised manuscript.
Reviewers’ comments:
Reviewer #1:
This is a very nice and thorough review on JCV and CNS oncogenesis written by one of the leading laboratories on the topic. I particularly liked the summarized tables separating all the different aspects of JCV interaction with multiple proteins, pathways, etc. There were only occasional typographical errors that should be checked.
>1. Page 4, 2nd paragraph, line 3: should read antibodies against JC”
Response: That is corrected
>2. Page 4, 2nd paragraph, line 8: Sentence “Genotyping analysis based… within the family” is confusing please clarify.
Response: The sentence is changed to clarify the point.
>3. Page 7, 2nd paragraph, line “c-Myc and cyclin D TCF promoter…” there may be a missing comma
Response: That is corrected
>4. Page 8, 1st paragraph, line *: “in this case, it reported is confusing, please clarify
Response: The sentence is changed to clarify the point
>5. Page 9, 1st paragraph, line 6: should read” JCv with human CNS tumors”. Same page last sentence, “showing that 57% to 38% 0f tumors…” should it include the word “respectively”. Please add if needed.
Response: That is corrected.
>6. Page 10, 1st paragraph, line 1: authors instead of Authors
Response: That is corrected.
>7. Page 12, 2nd paragraph, last line: monkey should read “monkeys”
Response: That is corrected.
>8. Please make sure genes are italicized appropriately in all text. Also, double-check abbreviations as some of these are not followed in the text. Also, review the name of proteins, “Tag” instead of Tag in a few instances. Maintain consistency either as all sentences saying B-lymphocytes or B lymphocytes.
Response: That is corrected
>9. The conclusion needs to be shortened. It also sounds repetitive from what has been mentioned already in the text
Response: The point is well taken, and we accordingly have shortened the discussion.
Reviewer 2 Report
This is a comprehensive review of mechanistic, animal study, and human sample evidence that JC virus (JCV) may play a role in the likely multifactorial causation of brain tumors. While there is scattered evidence to support the possibility that certain JCV strains may contribute to the transformation of CNS resident cells this concept is complicated by the almost ubiquitous presence of JCV and the fact that many of the brain tumors with evidence of JCV infection have been from individuals with PML or other immune perturbations. This, however, does not detract from a possible mechanistic contribution to tumorigenesis. The review is generally well written but suffers somewhat from cohesiveness. While the structure is generally sound it may be more effective to have the section on animal studies placed before the human data. References to animal studies have only limited bearing on the human data but may help substantiate mechanistic studies. Data from animal, in vitro and human studies need to be clearly separated and qualified. Evidence of human infection and evidence that the infection may drive cell transformation need to be clearly distinguished. There are also a variety of rather vague statements throughout. For example the term “some” is overused when referring to published findings. The number of studies, strain of virus, or number of samples should be specified rather than stating “some”. Discussions of JCV virus strains are also confusing. Most publications refer to archetypal strains rather than a specific archetype strain and there is little evidence that I can see that an “Archetype form converts to neurotropic strains such as Mad-1). They appear from the literature to consist of unique strains. Other references have been overstated. For example, the fact that JCV can be detected in urine samples only suggests that urine may be a mode of transmission, not that it is the major source of virus transmission. In addition, the data that B cells may carry virus into the brain in immunocompromised individuals may not pertain to the spread of the virus to the CNS in normal individuals. In the end there are major questions not really tackled in this review: How important might JCV infection be in the transformation of CNS resident cells in individuals that are not immunocompromised? Is a small amount of JCV DNA merely a biomarker of cell transformation or an active player in the process?
Author Response
Manuscript ID: ijms-857066.
The role of JC Virus in central Nervous System Tumorigenesis
Journal: IJMS
Corresponding Author: Hassen S. Wollebo
Date 8/22/20
We thank the editor and reviewers for reading our manuscript and providing helpful comments. Our detailed response is included below. Appropriate changes have been made to the text of the manuscript to address the points that were raised and incorporated into the revised manuscript.
Dear Dr. Wollebo,
Reviewers have now commented on your paper. You will see that they are advising that you revise your manuscript.
When revising your work, please submit a list of changes or a rebuttal against each point, which is being raised when you submit the revised manuscript.
Reviewers’ comments:
Reviewer #2:
>1. The review is generally well written but suffers somewhat from cohesiveness. While the structure is generally sound it may be more effective to have the section on animal studies placed before the human data.
Response: The point is well taken, and such a change is made
>2. Title: Data from animals in vitro and human studies need to be clearly separated and qualified.
Response: we have clearly summarize the human and animal studies in Table 2 and 3, respectively.
>2. Evidence of human infection and evidence that the infection may drive cell transformation need to clarify distinguished
Response: Regarding the evidence of human infection by JC virus, it is written in our abstract and introduction. JC Virus is a human neurotropic Polyomavirus belongs to the family Polyomaviridae and, it is the causative agent of progressive multifocal leukoencephalopathy (PML), which is a fatal neurodegenerative disease. Sero-epidemiological studies have indicated JCV infection is prevalent in the population (85%) and initial infection usually occurs during childhood. JCV's role to transform cells to cause tumors is well documented in animal models. However, there is no clear association between JCV presence in CNS and its ability to initiate CNS cancer and tumor formation in humans. This shortcoming in establishing the link between JCV and CNS tumorigenesis may be due to insufficient sample examined and the basic limitation of the traditional viral screening methods. The application of the new technology such as next-generation sequencing (NGS) which has high sensitivity and specificity to assess all exogenous agents within cancer samples and identified those pathways involved will significantly advance our understanding the role of JC Virus in CNS tumorigenesis
>3. There are also a variety of rather vague statements throughout. For example, the term “some” is overused when referring to published findings.
Response: We have only used five times the word “some” throughout the manuscript. We also made the necessary change to reflect the exact number of samples examined.
>4. The number of studies, strains of the virus, or number of samples should be specified rather than stating “some”
Response: We have nicely summarized the kind of studies and strain of the virus used in Table one.
>5. Discussions of JCV virus strains are also confusing. Most publications refer to archetypal strains rather than a specific archetype strain and there is little evidence that I can see that an “archetype form converts to neurotropic strains such as Mad-1.
Response: Two different forms of JCV exist based on the structure of the NCCR, the archetype and the neurotropic or prototypical form, often referred to as PML-type or Mad-1-like type, found in PML. The archetype (CY) has a simple highly conserved NCCR structure and may be the “wild-type” that is transmitted between individuals since it is the most abundant form in the environment. Neurotropic forms have deletions, duplications, and rearrangements in the NCCR compared to the archetype and have more variability in sequence (Mad-1, Mad-4, GS/B, Her1, Tokyo-1, etc) and possibly arise within an individual by neuroadaptation although direct evidence for this is lacking. Since the prototype has deletions and duplications compared to archetype, it seems reasonable to assume that if any interconversion occurs, it is in the direction of archetype to prototype. However, other models can be conceived for the relationship between archetype and prototype, e.g., rare quasi-species of the virus may exist and the abundance of these might vary depending on circumstances and selective pressures placed on the virus
>6. Other references have been overstated. For example, the fact that JCV can be detected in urine samples only suggests that urine may be a mode of transmission, not that it is the major source of virus transmission.
Response: This point is well taken, and the sentence is corrected accordingly.
>7. Data that B cells may carry the virus into the brain in immunocompromised individuals may not pertain to the spread of the virus to CNS in normal individuals.
Response: After primary infection, JCV remains in the body in a latent or persistent infection in the kidney, tonsils, bone marrow, spleen, brain, and lymph nodes. However, there are many critical areas where our understanding of JCV biology is incomplete. Specifically, the mechanism by which JCV reaches the brain. Currently, there is no widely accepted model for such a mechanism. According to one model, the immune cells especially B lymphocytes (Berger JR et al., 1997, Major EO et al., 2010) not only carrying PML-type JCV as a latent virus but also carry the virus around the body and enter the brain where the virus reactivates under immunosuppression. However, available evidence shows that neurotropic virus may exist, on occasion, in the bone marrow and brain of healthy individuals Furthermore, the notion that archetype undergoes conversion to the neurotropic form in the bone marrow especially in B cells still lacks direct support.
>8. How important might JCV infection be in the transformation of CNS resident cells in individuals that are not immunocompromised? Is a small amount of JCV DNA merely a biomarker of cell transformation or an active player in the process?
Response. careful examination of the available evidence reveals that neurotropic virus may exist, on occasion, in the bone marrow and brain of normal individuals who do not develop JCV associated pathogenesis. It is now widely accepted the role of immunosurveillance mediated by cytotoxic CD8+ T cells to eliminate infected cells and control virus dissemination in normal individuals. Despite the development of elaborate proposed models on how JCV reaches the brain, the notion that JCV reaches the brain, infect, and transform CNS resident cells still lacks direct support. It is our opinion that the presence of JC virus DNA or the viral proteins cannot serve as biomarkers for cellular transformation but the presence of JCV associated transcript in CNS tumor tissue might likely have a role in JCV associated tumorigenesis.